# Heat Stress in Broiler Chickens and the Effect of Dietary Polyphenols, with Special Reference to Willow (*Salix* spp.) Bark Supplements—A Review

**DOI:** 10.3390/antiox10050686

**Published:** 2021-04-27

**Authors:** Mihaela Saracila, Tatiana Dumitra Panaite, Camelia Puia Papuc, Rodica Diana Criste

**Affiliations:** 1National Research-Development Institute for Animal Biology and Nutrition (IBNA), Calea Bucuresti, 1, Balotesti, 077015 Ilfov, Romania; tatiana.panaite@ibna.ro (T.D.P.); criste.rodica@ibna.ro (R.D.C.); 2Faculty of Animal Production Engineering and Management, University of Agronomic Sciences and Veterinary Medicine of Bucharest, 59 Marasti Blvd, District 1, 011464 Bucharest, Romania; 3Faculty of Veterinary Medicine, University of Agronomic Sciences and Veterinary Medicine, 105 Splaiul Independentei, 050097 Bucharest, Romania; cameliapapuc@fmvb.ro; 4Academy of Romanian Scientists (AOSR), 54 Splaiul Independentei, 050094 Bucharest, Romania

**Keywords:** willow bark, chemical characterization, mechanism, broiler diet, heat stress

## Abstract

Over the last decade, there has been a growing interest in the use of a wide range of phytoadditives to counteract the harmful effects of heat stress in poultry. Willow (*Salix* spp.) is a tree with a long history. Among various forms, willow bark is an important natural source of salicin, β-O-glucoside of saligenin, but also of polyphenols (flavonoids and condensed tannins) with antioxidant, antimicrobial, and anti-inflammatory activity. In light of this, the current review presents some literature data aiming to: (1) describe the relationship between heat stress and oxidative stress in broilers, (2) present or summarize literature data on the chemical composition of *Salix* species, (3) summarize the mechanisms of action of willow bark in heat-stressed broilers, and (4) present different biological effects of the extract of *Salix* species in different experimental models.

## 1. Introduction

Heat is a real challenge in the poultry sector and a rising issue for many researchers on global warming and food safety. An exposure to a temperature above 30 °C means severe stress for broiler chickens [1]. Heat stress (HS) is one of the most common stressors that affect the production criteria in poultry [2,3]. Heat stress can be classified into two classes: acute HS, implying a short and rapid increase in temperature; and chronic HS, referring to extended exposure to high temperature [4]. Currently, climate change and increases in ambient temperature have been recorded in many regions [5]. Frequent excessive heat is now a problem in both hot climates and temperate countries. Furthermore, due to the higher tissue metabolism caused by intensive genetic selection for more rapid growth, commercial broiler has recorded a reduction in heat tolerance [6]. The scientific literature is abundant in evidence proving that HS affects performance [7,8,9], biochemical parameters [10,11] gut microbiota [12,13,14], immune response [15,16], and carcass quality and safety [17,18] of broiler chickens.

It was predicted considerable economic losses in several agricultural industries due to heat stress if urgent measures are not required [19]. Given the negative consequences of HS, tackling them has quickly become a particular point of interest in the animal breeding sector [20]. Nutrition management and feeding strategies were seen as more economically viable compared to non-nutritional strategies (e.g., hall cooling equipment) used to mitigate HS in chickens and to reduce losses [21]. For a long time, traditional antibiotics have been administered intensively to animals to prevent disease or as growth promoters [22], with some experts estimating that global consumption of antimicrobials in animals is twice that of humans [23]. However, this intensive use has led to the resistance of pathogenic bacteria, with significant public health implications such as bacterial infections with quinolones resistant bacteria (*Campylobacter* and *Salmonella*). Based on this concern, the EU Commission [24] decided to ban the use of antibiotics as growth promoters in feed. Nevertheless, according to Van Boeckel et al. [22], the global consumption of the antimicrobials in food animals (2010–2030) is expected to increase by 67%. Hence, there has been a great interest to develop novel alternative growth promoters to hinder the further growth of the antimicrobial resistance in animals and humans and to minimize the poultry growth inhibition caused by heat stress, to gain profitable production for the poultry industry. However, a promising alternative method is the supplementation with plants (phytoadditives) containing bioactive compounds that reduce the negative impact of HS. In this context, a nutritional solution is proposed; it includes a plant widely spread around the world, but which has not been given attention as a possible feed additive in the diet of heat-stressed broilers.

Genus *Salix* includes a number of species of trees and shrubs allowed to be used in a wider therapeutic application [25]. Willow bark is listed in the European Pharmacopoeia [26] and due to its salicin content (precursor of aspirin), it has been traditionally used for treating fever, pain, and inflammation in humans. The bioactive phytomolecules, medicinal values, and nutritional properties of this plant have been extensively studied by previous researchers [27,28,29,30]. *Salix* bark is a rich and inexpensive source of phenolic glycosides [31] and polyphenols, such as flavonoids and condensed tannins [32]. These constituents are reported to possess antirheumatic, antipyretic, anti-inflammatory, hypoglycemic [33], antibacterial [34], and antioxidant activities [30,35]. Although willow has been extensively studied in vitro and its undeniable beneficial properties have been proven, few studies are found on its applications in animal nutrition. This is all the more important since in recent years, for the safety of food and consumers, research has focused on finding natural alternatives to synthetic substances administered in the animal diet.

Given the negative consequences of HS on poultry, the interest in using natural feed additives in broiler diet and the above-mentioned properties of willow bark, this paper tries to summarize evidence of the potential use of willow bark for mitigating HS in poultry production. Thus, the review attempts to highlight both the properties of willow bark given by the bioactive compounds contained and its applications in the nutrition of broilers exposed to a stress factor, namely, heat stress.

## 2. Heat Stress and Oxidative Stress in Broiler Chickens

In the last decade, there has been an increase in the number of discussions about HS as a factor in inducing oxidative stress in chickens [4,36,37,38]. Oxidative stress in chickens is currently a topic of interest for several reasons. Firstly, it is related to a number of pathologies that affect the growth of birds [39]. A second consideration is that it alters the quality and safety of chicken meat. For example, degradation reactions in tissues affect muscle protein functionality and the sensory, nutritional, and shelf life of animal products [40,41,42].

### 2.1. Reactive Species Involved in Oxidative Stress

Mainly responsible for generating oxidative stress are the reactive species (RS), which can be classified into free radicals, non-radicals, and redox active transition metal ions. The free radicals hold one or more unpaired electrons, so they are chemically unstable. Thus, their main goal is to become stable. For this, they either donate the unpaired electron to another molecule or take an electron to obtain a stable configuration [43]. The free radicals include superoxide anion (O_2_^●−^), hydroxyl radical (HO^●^), nitric oxide (^●^NO), and nitrogen dioxide (^●^NO_2_). The non-radical reactive species are not free radicals but can easily lead to free radical reactions in living organisms. The non-radical reactive species include hydrogen peroxide (H_2_O_2_), singlet oxygen (^1^O_2_), ozone (O_3_), nitroxide anion (NO^−^), and peroxynitrite (ONOO^−^). The transition metals ions with two or more valence states may initiate HO^●^ generation in the Fenton/Fenton like reaction. The most aggressive metal ions involved in oxidative stress are Fe^2+^/Fe^3+^ and Cu^+^/Cu^2+^.

The reactive species, free radicals, and non-radicals are strongly reactive and may be grouped into reactive oxygen species (ROS) and reactive nitrogen species (RNS). Under normal physiological conditions, the ROS are formed as a consequence of the partial reduction of molecular oxygen [44]. ROS include the free radicals (RO^●^, ROO^●^, O_2_^●−^, HO^●^), and non-radicals (H_2_O_2_, ^1^O_2_, O_3_), able to produce several oxidation products [44]. Among them, the hydroxyl radical is the most reactive form of ROS produced by hydrogen peroxide via Fenton’s reaction [44]. Reactive nitrogen species include free radicals ^●^NO and ^●^NO_2_, and the non-radicals NO^−^ and ONOO^−^.

Oxidative stress occurs when RS are produced in excess, and the antioxidant defense systems of the chickens can be overwhelmed as the activity of antioxidant enzymes (superoxide dismutase, catalase, and glutathione peroxidase) decreases [45,46]. In other words, there is an imbalance between the pro-oxidants and the antioxidant system of the chicks. It was showed that oxidative stress can alter the redox balance of several cellular redox couples leading to the altered expression of key enzymes in detoxification, antioxidant defense, cell transposition, inflammatory responses, etc. [4]. For example, the exposure of broilers to 34 °C for 8 h per day has been reported to increase lipid peroxidation and decrease superoxide dismutase (SOD) and glutathione peroxidase (GPx) activities [47]. In contrast, broilers exposed to 38 ± 1 °C for 3 h showed increased catalase (CAT), SOD and GPx activities. This increase in the activities of antioxidant enzymes due to short-term exposure to HS has been considered to be a protective response to oxidative stress [48].

### 2.2. Effect of Heat Stress on Cellular Integrity. Biomarkers of Heat Stress

According to Ozcan and Ogun [49], the overproduction of ROS has been associated with cellular oxidative damage to DNA, lipid, and protein. DNA damage induced by oxidative stress includes base modifications, abasic sites, and strand breaks with effects in replication and transcription. The effects of HS on cellular integrity are depicted in Figure 1.

The phospholipid components of cell membranes are majorly rich in polyunsaturated fatty acids (PUFAs), which means that they are much more exposed to RS aggression [49]. Both ROS and RNS can contribute to lipid peroxidation, especially cell membrane phospholipids and lipoproteins [50]. Additionally, recent studies showed that the phospholipid bilayer becomes more disordered as a result of the oxidation products [51]. Mitochondria are the most affected sites in cells by the action of oxidative stress induced by HS. The explanation for this may arise from their implication in the essential functions of the cell, such as ATP production, in intracellular Ca^2+^ regulation, ROS production and scavenging, etc. [52]. Mitochondria are normally protected from oxidative damage by a mitochondrial antioxidant system [53], which consist of antioxidant enzymes such as SOD, CAT, GPx and glutathione reductase (GR) along with a number of micromolecular antioxidants, such as glutathione (GSH), ubiquinol (QH2) and vitamin E [54,55]. These protection systems control the production of free radicals, assuring a balance between antioxidants and prooxidants [56], but excessive ROS production in mitochondria leads to the damaging of proteins, lipids, and DNA, then decreases energy production efficiency and ultimately improves mitochondrial ROS production [4,38]. Higher ROS levels lead to significant mitochondrial dysfunctions and oxidative stress [4,38,57]. Studies in chickens have also shown that high temperatures induce both oxidation of lipids and proteins [58,59]. However, in another study assessing the effect of acute and chronic HS on chickens, no changes in both lipid and protein oxidation were observed [60]. What is certain is that depending on the type of HS applied to the chicks (i.e., acute or chronic HS), the deleterious modifications in the mitochondria are more intense or not. For example, exposing broiler chickens to acute HS results in excessive ROS production, as a consequence of increased oxidation levels of the mitochondrial substrate and electron transport chain activity [38]. In the second situation, when broiler chickens are exposed to chronic HS, some researchers [4] reported reductions in ATP synthesis and a higher prevalence of apoptosis or cell necrosis. Others [38] showed that chronic HS reduces the activity of antioxidant enzymes, consumes the body’s antioxidant reserves, and leads to an increase in ROS levels, which leads to oxidative stress.

In practice, there are some biomarkers evaluating oxidative stress. Malondialdehyde (MDA) is the main product resulting from the peroxidation of PUFAs. During the heat exposure of broilers, some authors [58] reported higher mitochondrial and plasma levels of MDA. Thus, the MDA level can be considered a biomarker for oxidative stress induced by HS. 4-Hydroxynonenal and acrolein are other aldehydes that are formed as a result of the lipid peroxidation of PUFAs.

Heat stress was also shown to cause protein oxidation [55], forming carbonyl groups, another biomarker of oxidative stress [41,61]. Carbonyl groups result from the oxidative break of peptide backbone [41,61], direct oxidation of amino acids (i.e., lysine, arginine, histidine, proline, glutamic acid, and threonine), or by the binding of aldehydes resulting from lipid peroxidation (4-hydroxynonenal or acrolein) [62].

### 2.3. Effect of Oxidative Stress on Inflammatory Response

Oxidative stress caused by HS is also correlated with the inflammatory response in chickens [47,63]. Effect of HS on inflammatory response is shown in Figure 2. Thus, the excessive growth of ROS induced by heat stress causes the activation of an inflammatory signaling cascade [64,65], which triggers pro-inflammatory cytokines such as interleukins (IL-1β, Il-6) and tumor necrosis factor α (TNF-α) production [66,67] by activating the pro-inflammatory transcription factor, the nuclear factor kappa-B (NF-κB) [68]. NF-κB is a transcription factor of major importance in inflammation, stress response, cell differentiation or proliferation, as well as cell death [43]. Inflammatory cells release a number of RS at the site of inflammation leading to amplified oxidative stress [69] and tissue damage [70]. Thus, inflammation and oxidative stress are interdependent pathophysiological processes [43,71]. For example, two situations are described below. In the first situation, if oxidative stress initially appears in an organ, inflammation will eventually develop and will further accentuate the oxidative stress. On the other hand, if the inflammation is the one that begins first, then oxidative stress will appear as a consequence that will further exacerbate the inflammation [72].

Normally, broilers are prone to colonization with various microorganisms because the intestines of the chicks are not mature. The typical reaction to an infection is localized inflammation of the gut, which occurs during the immune response to pathogens [73]. HS has been reported to lead to neuroimmune damage to broiler chickens, which further alters the intestinal-immune barrier by modifying the intestinal permeability, leaving pathogens free to migrate through the intestinal mucosa, thus generating an inflammatory response [50,74,75,76]. Numerous studies have shown that the inflammation of the intestine also decreases nutrient absorption and, therefore, there are significant reductions in chicken weight [50,63,77]. Inflammation in the gastrointestinal tract (GIT) is mediated by several stressors/infections that in turn generate ROS and disrupt the redox balance [50]. However, these changes are dangerous both in terms of poultry production and food safety.

## 3. Chemical Characterization of *Salix* spp. Bark

Several studies have been performed on the chemical composition of willow bark. Many compounds such as saligenin and its derivative salicin, flavonoids and tannins have been identified in the willow bark [32,78,79]. Salicin is considered the main active ingredient as it is metabolized to salicylic acid. All *Salix* species contain salicin, but in a low quantity, which is metabolized during absorption into various salicylate derivatives [80]. According to the European Pharmacopoeia (04/2008: 2312), the willow bark extract contains at least 5.0% of the total salicylic derivates, expressed as salicin (C_13_H_18_O_7_). The extract is obtained from the active principle of the plant, by an appropriate procedure, using either water, or an equivalent hydroalcoholic solvent with a concentration of maximum 80% ethanol *v*/*v*. Toxicity is far lower with willow bark than with aspirin due to the low levels of salicylates in the plant products [80]. However, willow bark is an important bitter tonic with marked astringent properties in humans, making it useful in chronic hypersecretory states, such as mucus discharges, passive hemorrhage, leucorrhea, humid asthma, diarrhea, and dysentery [80]. The effects of willow bark attributed to the salicin compounds include analgesic [78], anti-inflammatory [32,81,82], antipyretic [83,84], and antiplatelet activity [80]. These activities are also well known for supporting the body’s response to normal physiological stress [85]. Some researchers have reported that, besides salicylic derivatives, other substances like polyphenols (flavonoids, flavanols, and phenolic acids) can also contribute to the biological activities of willow bark [33,86]. Compared with the number of studies on the salicin content of willow bark and its biological effects, very few studies focused on its content of polyphenolic compounds. Table 1 reveals data regarding some saligenin derivatives (salicin, isosalicin, picein, salidroside, triandrin, salicoylsalicin, salicortin, isosalipurposide, salipurposide, and tremulacin) and polyphenols identified and quantified in different *Salix* bark species. Some researchers [32] reported a content up to 20% flavonoids and condensed tannins in willow bark. Among flavonoids, the most important ones are glycosides of naringenin, isosalipurposide and eriodyctiol [32]. There are many contradictory results in the literature regarding the content of antioxidant compounds in willow bark. These differences may be attributed to the *Salix* species, part of plant, solvents used, extraction method and time, environmental factors, etc.

Some researchers [79] extracted phytochemicals from the bark of different *Salix* species (*S. alba*, *S. babylonica*, *S. purpurea*, and *S. triandra*) in 70% ethanol. They found salicin in a concentration of 3.99 mg/g in *S. alba* (harvested in June) and phenolic compounds in a concentration of 6.16 mg/g. The same researchers [79] discovered in *S. alba* and *S. babylonica* the same classes of polyphenols (gallic acid, chlorogenic acid, p-hydroxybenzoic acid, syringic acid, epicatechin, p-coumaric acid, rutin, quercetin, trans-cinnamic acid, and naringenin), in close concentrations, as well as close concentrations of salicin (Table 1).

In *S. purpureea,* they did not find gallic acid, syringic acid, or p-hydroxybenzoic acid, rather they found caffeic acid and an approximately double amount of salicin. In *S. triandra*, some researchers [79] found gallic acid, chlorogenic acid, epicatechin, p-coumaric acid, rutin, quercetin, trans-cinnamic acid, naringenin, and salicin. It should be noted that, in *S. triandra*, the concentration of polyphenols and salicin were much lower than in the other species investigated. In the methanolic extracts of *S. alba* clone 1100 and *S. daphnoides* clone 1095, *S. acutifolia*, *S. daphnoides*, *S. purpurea* L., *S. triandra*, *S. herbacea*, *S. sachalinensis ‘Sekka’,* and *S. viminalis*, some authors [88] identified compounds such as vanillic, p-hydroxybenzoic and p-coumaric acids most frequently observed in the samples. However, the same researchers showed that the identified phenolic acids were in low concentrations, being less widespread than the glycosides or ester derivatives. It was reported that the solvent used for the extraction determined the selectivity of phenolic compounds [91]. Some researchers [89] showed that the ethanolic extracts of willow bark (*S. aegyptiaca*) are rich in polyphenols (18.39 mg/g), such as flavonoids, catechins, gallic acid, caffeic acid, vanillin, p-coumaric acid, myricetin, epigallocatechin gallate, rutin, quercetin, and salicin. In the wood of *S. nigra*, *S. babylonica,* and *S. eriocephala*, vanillic, syringic, ferulic, and p-hydroxybenzoic acids were identified, whereas in *S. caprea* wood, vanillic, 4-hydroxycinnamic, and ferulic acids were found [92]. Some authors [90] showed that the ethanolic extract of *S. daphnoides* bark contains naringenin-7-O-glucoside, isosalipurposide, salipurposide as phenolic compounds, whereas others [88] who followed a methanolic extraction of *S. daphnoides*, found other classes of phenolic compounds, such as p-hydroxybenzoic acid, vanillic acid, cinnamic acid, p-coumaric acid, ferulic acid, and caffeic acid.

Many researchers have reported major differences in the concentration of polyphenols of willow bark extracts [79,93,94]. Data regarding the antioxidant activity, total phenols, flavonoid and salicin content in different extracts of *Salix* species are presented in Table 2.

In the ethanolic extract of *S. alba* bark, some researchers [79] found a lower concentration of total phenols than those reported by others [35] who used boiled ethanol. It was showed that boiled ethanol has a better yield in extracting polyphenols (162 mg gallic acid equivalent/g) from *S. alba* bark [35]. The same researchers showed a dose-dependent inhibition of free radicals. For the same *Salix* species, *S. purpurea*, some authors [79] recorded a three times higher concentration of total phenols in the ethanolic extract than others [90] obtained when they used water. Some authors [89] showed that *S. aegyptiaca* bark had the highest antioxidant capacity when extracted in ethanol (169 ± 28 mg quercetin equivalents/g dried sample), followed by water (78 ± 4 mg quercetin equivalents/g dried sample) and cyclohexane (10 ± 0.1 mg quercetin equivalents/g dried sample. This could be explained by differences in the polarity of the solvents used for extraction.

Comparing the antioxidant capacity of several plant extracts, (white willow bark, rosehip, buckthorn, grape seeds, sesame, and willow buds), some researchers [99] showed that the hydroglyceroalcoholic extract of white willow bark has a free radical inhibitory capacity similar to that of grape seed and sesame seed extracts and greater than willow bud extract. Some authors [79] reported values that ranged from 1.83 to 7.79 µg/mL for IC_50_ DPPH^●^ in bark extracts of different species of *Salix* (Table 2). Moreover, the authors highlighted that the bark of *S. alba* had the highest radical scavenging activity against DPPH^●^, and *S. triandra* the lowest.

Due to the presence of these compounds, various in vitro studies have shown the antioxidant activity of willow extracts [25,100,101]. However, the direct antioxidant properties of willow bark extracts containing many polyphenols have attracted attention as the explanation for its successful in vivo application [98,102]. However, we believe that modern strategies for the administration of antioxidants with phenolic structures are needed, due to the different stability of these compounds in acidic, basic, or neutral environments. Thus, studies performed on green tea catechins (GTCs), which include (−)-epicatechin (EC), (−)-epicatechin gallate (ECG), (−)-epigallocatechin (EGC), and (−)-epigallocatechin gallate (EGCG) demonstrated that after oral administration of GTCs, polyphenols were partially absorbed due to the instability in the low alkaline environment of the intestine of EGCG and EGC [103]. A promising method of eluting the mechanisms of degradation of polyphenols in alkaline media is the use of co-encapsulation of these compounds in suitable gels. For example, co-encapsulation of EGCG in an emulsion gel containing sucrose and gel in the aqueous phase and polyglycerol polyricinoleate in the oil phase enhanced EGCG chemical stability under simulated gastrointestinal conditions and doubled its bioaccessibility [104].

## 4. Mechanisms of Action of Polyphenols

### 4.1. Polyphenols as RS Scavengers and ROS-Enzymes Synthesis Modulators

Due to the many properties and applications, polyphenols remain among the most intense molecules recently investigated by researchers in the field of human, animal, biological, and chemical nutrition. Polyphenols are a group of chemicals found in plants; they are characterized by (an)aromatic ring(s) bearing one or more hydroxyl (OH) groups. In a general classification, polyphenols are divided into four classes, including flavonoids, stilbenes, lignans, and phenolic acids [105]. Polyphenols are found in different parts of plants (leaves, bark, stems, roots, fruits, and flowers), so great attention was paid to them in the study of polyphenols.

Polyphenols are able to act on scavenging RS in several ways. For example, some authors [91] define two types of activities to scavenge ROS: direct activity of scavenging ROS or indirect activity by the induction of the synthesis of ROS-removing enzymes (e.g., SOD, CAT, etc.). Figure 3 shows these activities schematically. Polyphenols also have the ability to directly chelate transition metal ions, especially Fe^2+^ and Cu^2+^, ions that can generate highly reactive oxygen free radicals [106].

The antioxidant properties of polyphenols can be attributed to their chemical structure, including the presence of hydroxyl groups attached to the benzene ring, which are good hydrogen donors. In the case of flavonoids, it was reported that the B ring hydroxyl structure has a major role in the activity of free radical scavenging [107]. Polyphenols participate in the elimination of numerous ROS and RNS, such as hydroxyl radicals, peroxyl radicals, hypochlorous acids, superoxide anions, and peroxynitrite [108] by transferring the H atom from the OH group (polyphenols) or a single electron to the free radical or to a transition metal ion [41,91,109], as shown in Figure 3.

In the first of these cases, polyphenols (PhenOH) react with the free radical (R^●^) by transferring a hydrogen atom to it via the hemolytic cleavage of the O–H bond [110]. In hydrogen atom transfer mechanisms, the free radical eliminates an antioxidant hydrogen atom, and the antioxidant (e.g., polyphenol) itself becomes an oxidized radical, phenoxyl radical (PhenO^●^), which is more stable and less reactive than R^●^ [110,111]. In the case of the electron transfer, polyphenols can transfer an electron and thus reduce radicals, metal ions or various groups (e.g., carbonyl groups) [112]. The ability of the phenolic compounds to chelate redox active metal ions such as Fe^2+^, Cu^2+^, or Cu^+^ is imperative since these cations are able to catalyze the production of ROS, leading to lipid peroxidation, and DNA and protein damages. In the Fenton reaction, Fe^2+^ catalyzes, in the presence of hydrogen peroxide, the production of HO^●^, the most aggressive ROS. In the Haber–Weiss reaction, the superoxide radical (O_2_^●−^) reduces Fe^3+^ to Fe^2+^, which then are again involved in generating HO^●^ in the Fenton reaction [106]. Otherwise, Cu^2+^ oxidizes H_2_O_2_ to O_2_^●−^, resulting in Cu^+^. This cation reacts with excess H_2_O_2_ to produce HO^●^ via a Fenton-like reaction [113].

It has been established that polyphenol extraction solvents influence antioxidant capacity [114,115]. Some solvents are susceptible to the formation of hydrogen bonds with phenols and thus decrease antioxidant activity [116]. On the other hand, alcohols are those that have a double effect on the reaction rate between phenol and the peroxyl radical, acting as acceptors of hydrogen bonds.

In addition to the direct activity mentioned above, polyphenols can also exert antioxidant activity via the activation of the key transcription factor, nuclear factor (erythroid-derived 2)-like 2 (Nrf2). Polyphenols contribute to the separation of Keap1 protein (Kelch-like ECH-associated protein 1) from the Keap1-Nrf2 complex, by modifying the cysteine residues in Keap1 and subsequently leading to the translocation of Nrf2 into the nucleus. Hence, it binds to the ARE (antioxidant response element) sequence, leading to the expression of antioxidant enzymes [65,71,117], such as SOD, CAT, glutathione-S-transferase (GST), heme oxygenase 1 (HO1), etc. (Figure 3). The activation of the Nrf2-Keap1 pathway has been linked with the prevention or treatment of cancer and other chronic diseases such as diabetes [118]. According to some authors [119], the activation of NRf2 is considered one of the ways to reduce oxidative stress and inflammation. Others [120] showed that flavonoids may modulate the expression of γ-glutamylcysteine synthetase (γ-GCS), an important enzyme with implications in the antioxidant defense of the cellular system and in detoxification. Together with glutathione synthetase, γ-GCS is involved in the synthesis of GSH, a tripeptide thiol, with a pivotal role against ROS and RNS [121,122]. Some authors [123] reported that flavonoids stimulated the transcription of a critical gene for GSH synthesis in cells.

Given the diversity of polyphenols, there are a number of studies on the antioxidant mechanism of different classes of polyphenols. The most studied polyphenols as Nrf2 activators are quercetin, curcumin, and epigallocatechin-3-gallate (EGCG), etc. According to some researchers [124], curcumin possess the ability to increase the activity of GSH-linked detoxifying enzymes such as GSTs, GPx, and γ-GCS. Other authors [125] showed that EGCG is involved in the protection of neurons against oxidative stress by HO-1 activation via ARE/Nrf2 pathway and by the induction of HO1. However, it was reported [126] that black tea polyphenols stimulated NQO1 [NAD(P)H:quinone oxidoreductase 1] and GST in the liver and lung of mice via Nrf2-ARE. In vivo, tea polyphenols led to an increase in serum CAT, GPx and SOD activity and lowered the level of MDA.

These above-mentioned abilities of polyphenols led to many applications on animal models in order to investigate their clinical uses.

### 4.2. Polyphenols as Modulator of the Gut Microbial Balance

A balanced gut microflora is crucial for the host’s health [127]. Some researchers [128] reported that the composition of the microflora varies with the antibiotic administration, nutrition manipulation, etc. Some ingested bioactive compounds may influence the microbial populations [129]. Over time, polyphenols have been recognized as antimicrobial agents and acting as probiotics, stimulating the growth of beneficial bacteria (e.g., *Lactobacillus*, and *Bifidobacterium*) [130]. The beneficial effect of polyphenols highly depends on their bioavailability. Polyphenols are characterized by poor absorption, pass through the small intestine without being absorbed, remaining more in the large intestine, where they can influence the composition of the microflora [131]. Polyphenols are transformed by the intestinal microbiota in their metabolites (simpler phenolic compounds), which increases the bioavailability of polyphenols. The resulting metabolites are reported to have a higher biological activity than their precursor structures [132]. Some authors [133] showed that phenolic compounds contained in willow bark extract are metabolized by the gut bacteria to small phenolic metabolites like hydroxyphenylpropionic acids, hydroxyphenylvaleric acids, and salicylic acid. The mechanisms by which polyphenols exert the antimicrobial effect in the intestine is by decreasing the abundance of harmful bacteria by inhibiting the adhesion to intestinal epithelial cells and by reducing intracellular invasion and colonization by harmful bacteria [134]. Moreover, polyphenols promote the growth of the beneficial bacteria colonizing the intestine, which contributes to the gut barrier protection [135]. Therefore, the interactions between polyphenols and gut microflora can lead to the promotion of intestinal health.

There are limited studies evaluating the impact of willow bark polyphenols on the gut intestinal microbiota. Some authors [129] evaluated the influence of willow bark extract on human fecal microbiota, incubating the feces with 0 mg/mL (PBS and vehicle control), 2 mg/mL, and 10 mg/mL of willow bark extract at 0.5, 4, and 24 h. The authors reported that constituents of willow bark extract were metabolized by fecal bacteria and influenced the microflora composition; the beneficial microorganisms such as *Bacteroides* sp., *Parabacteroides* populations have increased.

Some authors [136] investigated the antibacterial effect of *Salix babylonica* L. hydroalcoholic extract against some pathogenic bacteria (*Escherichia coli*, *Staphylococcus aureus,* and *Listeria monocytogenes*). The authors showed antibacterial activity, mostly against Gram-positive bacteria *Staphylococcus aureus* and *Listeria monocytogenes*. The same researchers [136] demonstrated that the polyphenols contained by the *Salix babylonica* hydroalcoholic extract can be used as an alternative treatment against these microorganisms and, by this mechanism, they contribute to animal and human health.

If the anti-inflammatory and antioxidant activities of willow bark constituents are undeniable, its antimicrobial activity is a new perspective that can broaden the scope of willow bark. Therefore, this topic signals a critical lack of knowledge that needs to be filled due to its implications in understanding the interaction between willow bark polyphenols and microbiota and its practical applications.

### 4.3. Salicin and Salicin Derivatives: Analgesic, Antipyretic, and Anti-Inflammatory Effects

Although willow bark has been studied a lot, there were no clear data about the mechanism of action. Some studies showed that the properties of willow bark are due to its salicin content, a weaker precursor of aspirin [83]. Having antipyretic and analgesic effects, salicin has been studied as treatment of fever and diseases (like arthritis) in humans [83]. Salicin is closely related to aspirin and has a very similar action in the body [96]. When ingested, salicin, the active glycoside of willow bark, is hydrolyzed in the intestine to saligenin. Saligenin is absorbed and then oxidized to the therapeutically active compound salicylic acid, in the liver (Figure 4).

Salicylic acid inhibits cyclooxygenase 1 (COX-1) and cyclooxygenase 2 (COX-2), the same enzymes targeted by synthetic non-steroidal anti-inflammatory drugs (NSAIDs) to alleviate pain and inflammation [137]. Because of this conversion process, white willow generally takes longer to act than aspirin, but the effects last for an extended period of time [96].

On the other hand, some researchers [86] highlighted that the serum salicylate levels produced after the ingestion of willow bark extract are too low to explain its anti-inflammatory activity; it has been suggested that other constituents, such as tannins, flavonoids, or salicin esters, may contribute to the overall effect. The above-mentioned idea is also supported by others [32] in an in vitro study. In order to achieve this, a high bioavailability of phenolic compounds is needed, because a high content and the activity of the phytochemicals determined in samples did not lead always to a high activity in vivo. For this reason, attention should be given to the bioaccessibility and bioavailability of bioactive compounds. Some researchers [138] conducted an in vitro digestion study for the phenolic compounds identified in *Salix* bark. They found a high bioaccessibility and bioavailability of *Salix* bark phenolic compounds. For flavonoids, the authors indicated low bioavailability. This hypothesis is supported by some researchers [139], who explained that a wide range of flavonoids such as myricetin, kaempferol, quercetin, rutin, and luteolin possess immunomodulatory and anti-inflammatory activities, by inhibiting pro-inflammatory cytokine production and their receptors. Moreover, some authors [89] reported a considerable content of myricetin, rutin, and catechin in willow extracts and found that those compounds potentially contribute to the anti-inflammatory functions of willow extracts.

## 5. Biological Effects of *Salix* Species Extracts in Different Experimental Models

Table 3 shows a selection of literature data on the in vitro effects of the bark of *Salix* species. Effects such as anti-inflammatory, antioxidant, antibacterial, antifungal, and anti-proliferative ones have been proven.

In an in vitro model of human umbilical vein endothelial cells, some authors [140] demonstrated that the antioxidant activity of willow bark is due to the presence of phenolic compounds and not just salicin. This was evidenced by the fact that the extract that did not include salicin activated ARE-luciferase activity. The antioxidant response element is responsible for encoding antioxidant and cytoprotective detoxifying enzymes and proteins, playing a pivotal role in redox homeostasis [144,145]. Nrf2 is the primary transcription factor that binds to the ARE, and through heterodimerization with other leucine-zipper with transcription factors, it activates the expression of the antioxidant enzymes genes [146]. These observations were confirmed in other experiment [140] showing that the phenolic components of willow bark activated the Nrf2 pathway, thus inducing the dose-dependent expression of phase II defense enzymes such as heme oxygenase1 (HO1), g-glutamyl cysteine ligase (GCL), NQO1 and increases the intracellular GSH. Through those actions, willow bark can overcome oxidative stress. The suppression of oxidative stress by inducing antioxidant enzymes of willow bark extract has been confirmed also by others [25,33].

Some researchers [141] studied the anti-inflammatory mode of action of willow bark in primary canine articular chondrocytes treated with interleukin (IL)-1β), a pro-inflammatory cytokine. The willow bark extract expressed a reduction in IL-1β, COX-2 and matrix metalloproteinases (MMP-9 and MMP-13) expression via NF-κB inhibition. NF-κB is the key mediator of inflammation, normally found in the inactive state. Under the action of inflammatory stimuli (cytokines, oxidants, bacteria, and viruses), inhibitory proteins that bind NF-κB in the cytosol are released [65]. Therefore, free NF-κB migrates into the nucleus, which ultimately leads to the transcription of pro-inflammatory genes and the increase of pro-inflammatory cytokines (e.g., IL-1β, IL-3, IL-6, TNF-α). Polyphenols have been reported to act as inhibitors of NF-κB [147,148]. Few studies showed that the anti-inflammatory properties of willow bark are associated with the inhibition of cyclooxygenases and pro-inflammatory cytokines [81,149].

The anti-inflammatory effects of willow bark extracts (from *S. daphnoides*, *S. purpurea,* and *S. fragilis*) on human monocytes were also reported [142]. The authors showed that the willow bark extract can inhibit the inflammatory cytokines such as gamma interferon (IFN-γ), and lipopolysaccharide (LPS) in human monocytes. The LPS has the potential to activate multiple inflammatory pathways, including the up-regulation of TNF-α, one of the most powerful pro-inflammatory agents [150]. Moreover, the same researchers have reported that IFN-γ could amplify the effects of LPS. In addition, others [142] recorded a significant reduction of nitrite and NO release (inflammatory mediators) in human monocytes.

Similarly, in human monocytes, some authors [82] showed that the *Salix* extract 1520 L inhibited LPS-induced IL-1β and IL-6 release, but not salicin and salicylate. They also reported that, compared with salicin and salicylate, *Salix* extract 1520 L inhibits COX-2-mediated prostaglandin E2 (PGE2) release; the conclusion was that this anti-inflammatory effect is attributed to other active compounds. 

Some authors [100] investigated the anti-proliferative effects of a standardized extract (STW 33-I) and a polyphenol-rich fraction of STW 33-I in comparison to the NSAIDs such as acetylsalicylic acid and diclofenac. The STW 33-I and its fraction E showed anti-proliferative and dose-dependent pro-apoptotic effects in HT 29 cells. Moreover, the willow extract stimulated only the mRNA expression of COX-1, whereas the NSAIDs inhibited both COX-1 and COX-2.

In addition to the above-mentioned effects, some researchers [35] reported the in vitro antibacterial action of the *S. alba* extract against *Staphylococcus aureus*, *Pseudomonas aeruginosa,* and *Candida albicans*. Moreover, the same authors showed a decrease in the viability of HL-60 cells, suggesting a potential implication in cancer prevention. Some authors [34] supported the antibacterial effect of *S. alba*, showing that, in liquid media, the willow extract had a 100% inhibitory effect against *Bacillus cereus* and *Staphylococcus aureus*, while on agar media, it was the most active antibacterial agent against *Listeria monocytogenes,* and *Bacillus cereus*. A researcher [80] highlighted that salicylates have unique physicochemical properties, caused by the close steric vicinity of the acetate hydroxyl group to the carboxyl group. The same authors explained that the major functional consequence is the action of salicylate as protonophore, for example, in mitochondrial membranes, to uncouple oxidative phosphorylation because of the abolition of the membrane impermeability to protons. Due to the presence of hydroxyl groups, the phenols have the capacity of incorporating into the lipid membranes, thus increasing their permeability, which makes the pathogen bacteria more sensitive to antibacterial compounds [151,152]. Some researchers [143] showed that volatile compounds in *S. acutifolia* (formic acid pentyl ester, hexanoic acid ethyl ester, pentanoic acid methyl ester, 3,5-octadien2-one, and 2-pentenal) reduced aflatoxin production of *A. parasiticus* (>90%), one of the main aflatoxin producers. 

## 6. Effects of Different Forms of Inclusion of *Salix* Species in the Diet of Heat-Stressed Broilers

As previously mentioned, it has been shown on numerous occasions that heat stress affects broiler performance and carcass quality. With respect to the application of dietary willow bark on performance of heat-stressed broilers, the literature is rather scarce. Some summarized data are presented in the Table 4.

Some researchers [153] highlighted that *Salix babylonica* could be used as a natural alternative to replace the synthetic acetylsalicylic acid (ASA) in broiler diet. Thus, they compared the effect of the extract of *S. babylonica* leaves with those of ASA. Under HS condition (35 °C), the authors showed similar performance between Arbor Acres broiler on a diet with ASA (100 mL/day of 0.1% ASA solution) and those on a diet supplemented with *S. babylonica* leaf extract (100 mL/day). Moreover, *S. babylonica* leaf extracts improved heat tolerance, feed intake, body gain, feed conversion rate, and reduced mortality of heat-stressed broilers. Acetylsalicylic acid, also known as aspirin, was previously studied as supplement in the diet or water of heat-stressed broilers and has been shown to attenuate the negative effects of high temperatures [156] or even to improve production performance and physiological traits [157,158]. However, in recent years, given the trend of using natural ingredients in animal feed, *Salix* species has gained attention as alternative to aspirin. Thus, some researchers [95] reported that the supplementation of dietary hydroalcoholic willow bark (*S. alba*) extract powder (25 and 50 g/kg diet) did not significantly affect the performance of broilers exposed to 32 °C (14–42 days). Similar results were obtained when some authors [99] used 1% hydroglyceroalcoholic extract of white willow bark (*S. alba*) in broiler diet (14–28 days) reared under HS. Under thermoneutral conditions, some researchers [159] found that the higher level of inclusion of *S. alba* bark in the diet (0.05%) significantly increased final body weight (+9.34%) and average daily gain (+10.76%) of Cobb 500 broiler (14–42 days) compared with the lower level of inclusion (0.025%). In this context, many researchers found a dose-dependent efficacy for *S. alba* bark [93,94,159]. Thus, it can be stated that *S. alba* bark improved broiler performance only under thermoneutral conditions. According to some authors [4], many phytochemicals had beneficial effects in heat-stressed poultry but were less or not effective in non-heat-stressed counterparts. In this context, an important role is played by the genetic characteristics of the bird, plant species, form of use, dose of inclusion, bioavailability, etc. 

Apart of those investigations, the growth promoting effect of *Salix* polyphenols on broiler chickens can also be considered a result of inhibiting pathogenic bacteria and stimulating bacteriocin-producing bacteria. It is well-known that diet has a critical role in modifying the microbiome, which has strongest influence on the feed efficiency in poultry [160]. Thus, there is a strong relationship between diet-gut microbiota- avian growth performances. Microbiota manipulation by providing nutrients, inhibits pathogen intestinal colonization (e.g., *Clostridium perfringens* and *Enterococcus* spp., *Campylobacter*, *Salmonella*, and *E. coli*), improves intestinal barrier function and growth performance such as body weight and feed conversion ratio [161]. Polyphenols from willow bark have low oral bioavailability, reach the colon, and afterwards being metabolized by gut microbiota [129] into derived metabolites which can affect microbial composition of the gut and signalling pathways [135]. For example, Lactobacilli may metabolize polyphenols providing energy for cells and simpler compounds that can interfere with metabolic activities of gut bacteria [160]. Several researchers highlighted that microbiota is specialized in the production of output metabolites mainly short chain fatty acids (SCFAs), that can be used as source of energy for the animal growth [162,163]. It was reported that phenolic acids such as chlorogenic acid, caffeic acid, rutin, and quercetin significantly increased the production of propionate and butyrate [164]. *Salix* polyphenols such as (−)-epigallocatechin gallate (EGCG) may produce SCFAs including acetic acid, propionic acid, butyric acid, which can modulate appetite and energy intake [165], with favourable effect on broiler performance [166,167]. Many studies revealed reciprocal interaction between phenolics and gut microbiota [135].

As we already mentioned, microbiota may metabolize polyphenols, but also polyphenols or its metabolites can modulate the microbiota by inhibiting pathogenic bacteria and stimulating populations of beneficial bacteria [128,135]. Flavonoids such as catechin, quercetin, naringenin, and phenolic acids exert antimicrobial activity against different pathogenic bacteria affecting broiler performance [168,169]. Dietary quercetin reduced the population of total aerobes and coliforms and increased the population of Bifidobacteria in laying hens [168], which led to an improved growth performance and feed efficiency [170]. Catechin and epigallocatechin from grape extract decreased the abundance of *Escherichia coli*, *Enterobacteriaceae* in broiler chickens [171]. Dietary supplementation with 400 ppm of quercetin stimulated the multiplication of Lactobacilli in broiler cecum [172]. Lactobacilli are able to produce lactic acid and proteolytic enzymes, which enhance nutrient digestion in the gut and improve weight gain in livestock animals [173,174]. Moreover, lactobacilli may decrease the colonization by enteropathogens through several mechanism such as competitive exclusion, antagonistic activity, and the production of bacteriocins [175]. However, it has been reported that an increase in ileal *L. salivarius* can reduce the growth performance of broilers due to its ability to deconjugate bile acids [176]. 

Although a large number of papers have been published on this topic, the relationship between polyphenols and gut microbiota-growth performance of chickens has not yet been fully clarified. There are researchers considering that are difficult to identify specific bacterial populations that could improve productivity and modulate the microbiota to a desired one [177]. Further investigation (e.g., omics advanced approaches) would be essential to elucidate the interaction between *Salix* metabolites in the diet, intestinal microbiota, and growth performance, clarifying those genes and microorganisms involved in the metabolization of polyphenols.

Some researchers [155] showed that feed conversion ratio and carcass weight were not affected by the supplementation of the chickens’ diet with *S. tetrasperma Roxb* leaf extract (50 and 100 mg/L of drinking water for 7 days in 4 h/day), either reared in HS (34 ± 1 °C) or not. In addition, the same authors reported that *Salix* extract supplementation in chicken diet decreased the weight of abdominal fat (−8.42%) only in chickens reared under normal temperature, while in heat-stressed broiler it had a negative effect, increasing the weight of abdominal fat (+18.18%). The clear mechanism of explaining these results is difficult to be stated and need to be further studied. It is thought that *Salix* spp. extract acts to counteract the negative effects of HS in broiler through its antioxidant activity [95,155]. 

One of the contributions of *Salix* spp. bark found in the literature on heat-stressed broilers is the decrease in serum cholesterol and glucose levels. It is well-known that high temperatures as a stress factor cause neuroendocrine and metabolic changes leading, among other things, to increases in blood glucose and cholesterol levels [178]. Some authors [95] reported significantly lower serum cholesterol (−26.29%) and triglycerides (−30.65%) in heat-stressed chickens fed with a diet supplemented with powder of *S. alba* bark (50 g/100 kg diet) compared to those on a conventional diet. Similarly, diet supplementation with 1% extract of *S. alba* bark led to a decrease in serum glycaemia (−16.35%), cholesterol (−11.25%), and triglycerides (−22.16%) of heat-stressed broiler, compared to those fed a conventional diet [154]. Another study performed on mice showed the effectiveness of *S. tetrasperma* extract in reducing the blood glucose concentration [179]. The results on the decrease in glucose levels were explained by the property of the phenolic compounds contained in *S. alba* in regulating glucose homeostasis and improving insulin sensitivity [180]. Many clinical studies highlighted that phenolic compounds could interact with the cholesterol carriers and transporters present across the brush border membrane [181,182], thus resulting in lower cholesterol absorption [183].

Polyphenols from *S. alba* bark were also found to protect the hepatic function of broilers both in HS and in thermoneutral conditions. The liver function is more susceptible to be affected by HS, due to its central role in maintaining the overall metabolism of the organism [184]. Worth mentioning is that the broilers subjected to HS (32 °C) and fed a diet with 1% extract of *Salix alba* bark had lower serum levels of alanine aminotransferase (ALT), aspartate aminotransferase (AST), and lactate dehydrogenase (LDH) than those fed a non-supplemented diet [154]. It is clearly stated that HS is involved in inducing oxidative stress in cells by accelerating the formation of ROS, and thus disturbing the equilibrium between ROS and antioxidant defense system [4,37]. This state expects an alteration of the antioxidant enzymes, increases the susceptibility of lipids to oxidation [185]. There are studies supporting that the bark extract of *Salix alba* might suppress oxidative stress by inducing antioxidant enzymes [94,159]. Some researchers [95] reported an improvement in the liver oxidative status of heat-stressed broilers (by decreasing the malondialdehyde level in the liver) as a consequence of diet supplementation with powder of *S. alba* bark (25 mg and 50 mg/100 kg diet). Similar results were found by others [159] for broilers under thermoneutral conditions. Moreover, they showed that dietary supplementation with polyphenolic powder of *S. alba* bark at 0.025% and 0.05% significantly reduced the lipid peroxidation by decreasing the level of malondialdehyde in broiler liver and reduced protein oxidation by decreasing the protein carbonyl groups compared with the non-supplementing diet. Moreover, they found that dietary supplementation with 0.05% *S. alba* bark increased the total antioxidant capacity and GSH levels in broiler liver tissue, without affecting SOD activity. Those effects were attributed by the cited authors to the polyphenolic content and antioxidant activity of *S. alba* bark. Phenols are involved in the antioxidant defense of organism, protecting cellular damage from the harmful effects of reactive oxygen species [186]. Some authors [140] explained that willow bark extract might induce antioxidant enzymes and prevent oxidative stress by the activation of Nrf2 independent of salicin. 

In addition to the above-mentioned effects, *Salix* spp. bark also contributes to maintain the balance of intestinal microflora of broilers reared under HS conditions. That achievement is all the more important as heat stress affects the composition of the intestinal microbiota of chicks, by decreasing the number of bacteria of the genus *Lactobacillus* and *Bifidobacterium* and increasing those of the genus *Clostridium* and total coliforms [185,187,188,189]. This causes an intestinal imbalance that generates many diseases in chickens, as well as poses a risk to the safety of chicken meat. It was stated that the bioactive compounds (e.g., polyphenols) of *Salix* spp. bark might reduce the multiplication of pathogenic bacteria and stimulate the development of the beneficial ones (e.g., lactobacilli). In this regard, some authors [154] reported that the dietary hydroglyceroalcoholic willow bark extract (1%) significantly decreased the *Enterobacteriaceae* (−6.36%), *E. coli* (−2.03%) and staphylococci (−8.22%) populations in broiler caecum at 42 days. Furthermore, some authors [95] found that adding 25 g and 50 g/100 kg diet powder of *S. alba* bark in the heat-stressed broiler diet the number of pathogenic bacteria (*E. coli*, staphylococci) decreased and that of lactobacilli increased in the broiler caecum at 35 and 42 days of age.

## 7. Conclusions

Heat is a real challenge in the poultry sector as it is inducing oxidative stress, associated with cellular oxidative damage and the inflammatory response. Many bioactive compounds with antioxidant, anti-inflammatory and antimicrobial activity, such as polyphenols and salicin, have been identified in *Salix* spp. extracts. Some studies have reported that the inclusion in the diet of heat-stressed broilers of supplements represented by extracts and powders obtained from the bark of *Salix* spp. has led to a reduction in the level of oxidative stress biomarkers, a decrease in pathogenic bacteria; and an increase in the number of lactobacilli in the caecum, of the final body weight, the average daily gain, and the average daily feed intake, as well as a decrease of the panting rate. 

Based on the reviewed data, it could be concluded that due to its antioxidant property, dietary willow bark might be an effective supplement to alleviate the adverse effect of heat stress on biochemical parameters, oxidative status, and gut microflora composition of heat-stressed broilers. Further studies would be helpful to adjust the optimum supplementation dose of willow bark in broiler diets and its effect on other parameters such as immune response, quality and safety of chicken meat, morphology, and development of organs and intestine of heat-stressed broilers. 

## Figures and Tables

**Figure 1 antioxidants-10-00686-f001:**
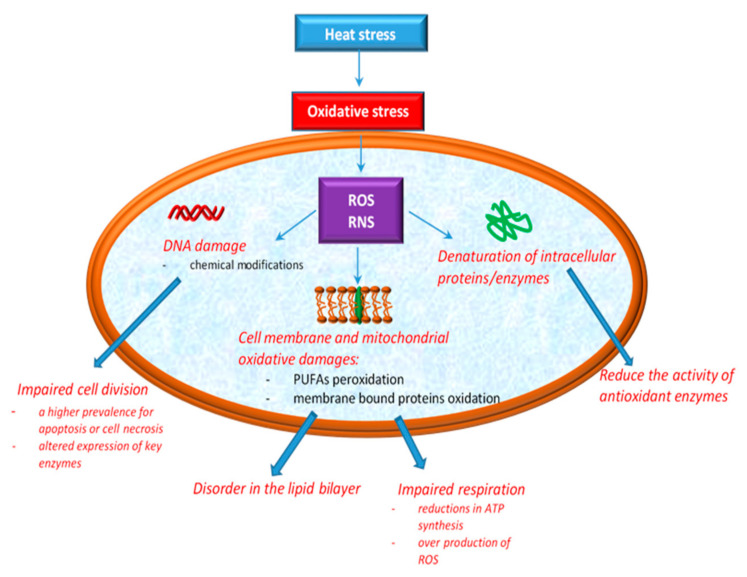
Cellular oxidative damage induced by heat stress.

**Figure 2 antioxidants-10-00686-f002:**
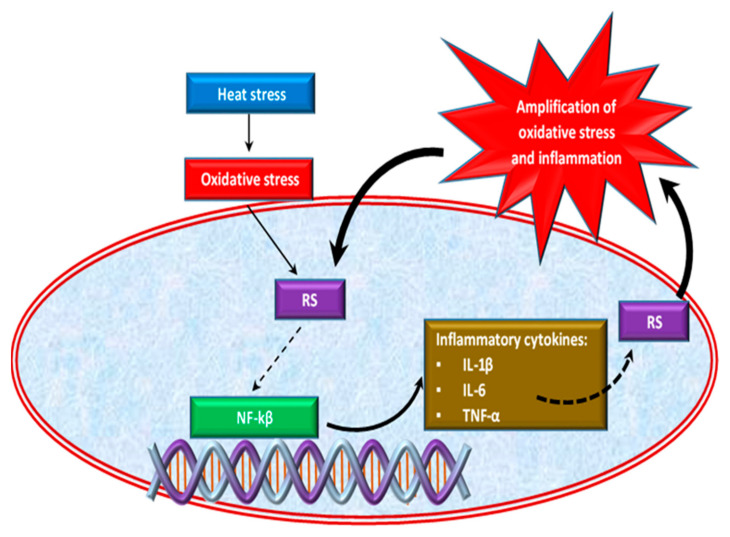
Effect of heat stress on inflammatory response.

**Figure 3 antioxidants-10-00686-f003:**
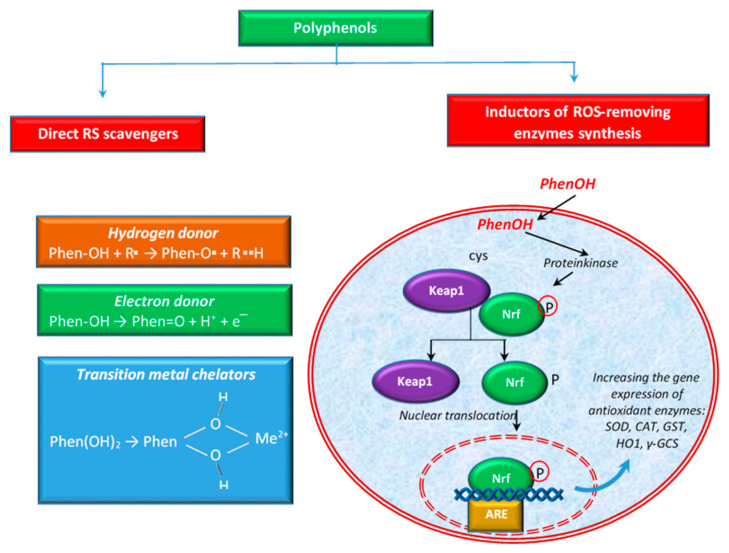
Different ways of polyphenols to eliminate ROS.

**Figure 4 antioxidants-10-00686-f004:**
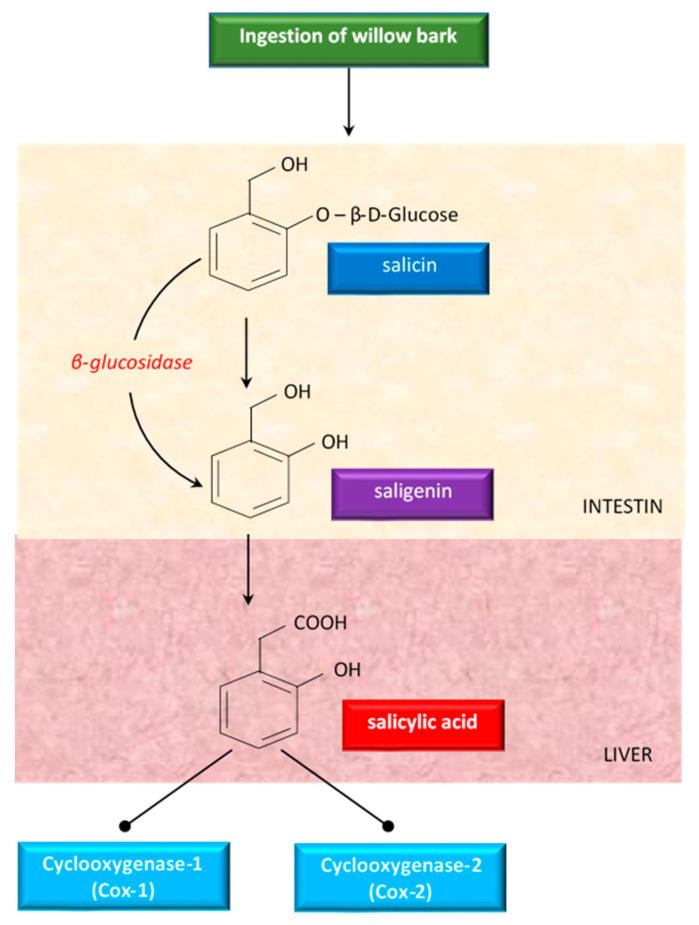
An overview of the metabolism of salicin derivatives from willow bark.

**Table 1 antioxidants-10-00686-t001:** Polyphenols and saligenin derivatives identified and/or quantified in different barks of *Salix* species.

Species	Origin and Sampling Period	Extraction Method	Compounds Detection Method	Identified/Quantified (mg/g) Compounds		Reference
				Polyphenols	Saligenin Derivatives	
*S. alba*	Pecenjevce (June)	70% Aqueous ethanol by maceration for 48 h at room temperature (25 °C)	HPLC with diode array detection	Gallic acid (0.17) Chlorogenic acid (1.65) p-Hydroxybenzoic acid (0.32) Syringic acid (0.22) Epicatechin (1.17) p-Coumaric acid (0.15) Rutin (1.75) Quercetin (0.38) trans-Cinnamic acid (0.15) Naringenin (0.20)	Salicin (3.99)	[79]
*S. babylonica*	Bosut riverside, Morovic (September)			Gallic acid (0.17) Chlorogenic acid (1.92) p-Hydroxybenzoic acid (1.21) Syringic acid (0.34) Epicatechin (2.68) p-Coumaric acid (0.15) Rutin (1.36) Quercetin (0.52) trans-Cinnamic acid (0.57) Naringenin (0.27)	Salicin (3.11)	
*S. purpurea*	Mountain Deli Jovan (August)			Chlorogenic acid (1.14) Caffeic acid (1.05) Epicatechin (2.08) p-Coumaric acid (1.53) Rutin (4.30) Quercetin (1.13)	Salicin (7.53)	
*S. purpurea*	Mountain Deli Jovan (August)	70% Aqueous ethanol by maceration for 48 h at room temperature (25 °C)	HPLC with diode array detection	trans-Cinnamic acid (0.13) Naringenin (0.26)		[79]
*S. triandra*	Vlasina Lake (July)			Gallic acid (0.26) Chlorogenic acid (1.63) Epicatechin (1.77) p-Coumaric acid (0.22) Rutin (1.73) Quercetin (0.67) trans-Cinnamic acid (0.53) Naringenin (0.33)	Salicin (2.87)	
*S. subserrata*	Sharkia, Egypt, (March)	Exhaustively extraction with methanol at room temperature	Silica gel column chromatography	(+) Catechin 1,2-Benzenedicarboxylic acid, Methyl 1-hydroxy-6- oxocyclohex-2-enecarboxylate Catechol propyl acetate	bis (2-ethylhexyl) ester saligenin	[87]
*S. alba* *S. daphnoides*	Willow cultures of the University of Warmia and Mazury (Olsztyn, Poland)	Exhaustively extraction with methanol (3 × 120 mL, 60 °C)	MGD-HPTL	p-Hydroxybenzoic acid Vanillic acid Cinnamic acid p-Coumaric acid Ferulic acid Caffeic acid		[88]
*S. acutifolia* *S. daphnoides*, *S. purpurea* L. *S. triandra*	From their natural habitat in west Poland (March)	Exhaustively extraction with methanol (3 × 120 mL, 60 °C)	MGD-HPTL	p-Hydroxybenzoic acid Vanillic acid Cinnamic acid p-Coumaric acid Ferulic acid Caffeic acid		[88]
*S. herbacea* *S. sachalinensis* *S. viminalis*	Garden of Medicinal Plants (Medical University of Gdańsk, Poland)					
*S. purpurea*	Labofarm (Starogard Gd., Poland, March)					
*S. aegyptiaca*	Ghaene ghom, Iran, (unknown season)	Ethanol extraction (1:10, *w*/*v*) by sonication 20 min	HPLC method with a PDA detector	Gallic acid (0.69) Caffeic acid (0.06) Vanillin (1.53) p-Coumaric acid (0.80) Myricetin (5.87) Catechin (0.93) Epigallocatechin gallate (2.39) Rutin (quercetin-3-rhamnosyl glucoside) (4.59) Quercetin (1.47)	-	[89]
*S. daphnoides*	Finzelberg, Andernach, Germany (unknown season)	Methanol extraction, 1:25 (*w*:*v*) by stiring for 2 h	RP-HPLC coupled to electrospray triple-quadrupole MS and MS/MS	Naringenin-7-O-glucoside Isosalipurposide Salipurposide	Saligenin, Salicylic acid Salicin Isosalicin Picein Salidroside Triandrin Salicoylsalicin Salicortin Tremulacin	[90]

**Table 2 antioxidants-10-00686-t002:** Total phenolics content, total flavonoids content, salicin content, and antioxidant activity of different bark extracts of *Salix* species.

Species	Origin and Sampling Period	Extraction Method	Total Phenolics (mg GAE/g ^1^)	Total Flavonoids (mg QE ^2^/g dw ^4^)	Salicin Content (mg/mL)	Antioxidant Activity	Reference
*S. alba*	Pecenjevce, June	70% Aqueous ethanol extraction by maceration for 48 h at room temperature (25 °C)	40.9	3.48	3.99 mg/g	IC_50_ DPPH ^3^ = 1.83 μg/mL	[79]
*S. babylonica*	Bosut riverside, Morovic, September		20.17	3.13	3.11 mg/g	IC_50_ DPPH ^3^ = 2.59 μg/mL	
*S. purpurea*	Mountain Deli Jovan, August		69.1	31	7.53 mg/g	IC_50_ DPPH ^3^ = 4.73 μg/mL	
*S. triandra*	Vlasina Lake, July		18.41	2.88	2.87 mg/g	IC_50_ DPPH ^3^ = 7.79 μg/mL	
*S. alba*	Plant Extract, Radaia, Cluj County, unknown season	Hydroalcoholic extraction in grain alcohol and water by maceration	4.67		98%	35.13 mM equivalent ascorbic acid 35.97 mM equivalent vitamin E	[95]
*S. alba*	Phyto concentrate India, unknown season	Methanol extraction by sonication for 30 min	-	-	1.92%	-	[96]
*S. purpurea*	cultivated on the sandy soil (heavy loamy sand) of experimental fields at the University of Life Sciences in Lublin, November	Water extraction by shaken for 60 min at room temperature	20.04	-	-	EC_50_ LPO ^5^ = 8.06 mg/mL	[93]
*S. myrsinifolia*			23.10	-	-	EC_50_ LPO ^5^ = 8.31 mg/mL	
*S. alba*	-	Soxhlet extraction with ethanol for 7 h	162	-	-	DPPH ^3^ In% = 12.50, 37.50 and 80.00% of 10, 50 and 100 μg/mL	[35]
*S. tetrasperma Roxb.*	Zagazig City, Sharkia Province, Egypt, unknown season	Methanol extraction	-	-	-	IC_50_ DPPH ^3^ = 94.5 µg/mL	[97]
*S. aegyptiaca*	Ghaene ghom, Iran, unknown season	Ethanol extraction (1:10, *w*/*v*) by sonication for 20 min.	212	479	3.1	169 mg QE ^2^/g dried sample	[89]
*S. aegyptiaca*		Water extraction (1:10, *w*/*v*) by sonication for 20 min.	139	243	0.07	78 mg QE ^2^/g dried sample	
*S. caprea*	Finnish origin, unknown season	80% Aqueous methanol extraction, using Ultra Turrax mixer, 1 min.	75.5	-	-	96% Inhibition (500 ppm) MLO ^5^	[98]
*S. alba*	Finnish origin, unknown season	80% Aqueous methanol extraction, using Ultra Turrax mixer, 1 min.	58.6	-	-	96% Inhibition (500 ppm) MLO ^5^	

^1^ Gallic acid equivalents. ^2^ Quercetin equivalents. ^3^ 2,2-Difenil-1-picrililhidrazil radical. ^4^ Catechin equivalents. ^5^ Methyl Linoleate Oxidation.

**Table 3 antioxidants-10-00686-t003:** In vitro studies on the effects of the bark of *Salix* species on different experimental models.

Species/Origin	Experimental Model	Effect	Reference
Willow bark extract (Shamanshop, Camden, NY, USA)	Human umbilical vein endothelial cells	—willow bark extract failed to activate ARE ^1^ —luciferase activity, whereas a salicin-free willow bark extract fraction had intensive activity —induced antioxidant enzymes and prevented oxidative stress through activation of Nrf2 ^2^ independent of salicin	[140]
*S. alba* (chloroform extract)	Primary canine articular chondrocytes	—anti-inflammatory and anabolic effects on chondrocytes, reducing cytokine induced activation and up regulation of pro-inflammatory enzymes (MMPs and COX-2 ^5^) and NF-*κ*B ^3^	[141]
5 fractions of a willow bark extract from *S. daphnoides*, *purpurea* and *fragilis*	Human monocytes	—↓ nitrite and NO ^7^ release —inhibited the inflammatory cytokines (IFN-γ ^9^), and lipopolysaccharide (LPS)	[142]
*Salix* extract 1520 L (ethanol extract)	Primary human monocytes	—inhibits COX-2 ^5^-mediated PGE_2_ ^8^ release through other compounds than salicin or salicylate	[82]
Willow bark extract STW 33-I (water extract) and a polyphenol-rich fraction of STW 33-I	Colon-carcinoma cell line HT-29	—anti-proliferative and pro- apoptotic effects on HT-29 —inhibited COX-1 ^6^ expression	[100]
*S. alba* (ethanol extract)	Bacterial strains and one yeast HL-60 ^4^ cells	—significant antioxidant activity and antimicrobial activities against *Staphylococcus aureus*, *Pseudomonas aeruginosa* and *Candida albicans* —highly cytotoxic to HL-60 cells, dependently on the dose and time	[35]
*S. alba* (water extract)	Bacterial strains	—antimicrobial activity against *E. coli, Staphylococcus aureus*, *Listeria monocytogenes*, *Bacillus cereus* and *Salmonella enteritis*	[34]
*S. acutifolia* (ethanol extract)	*A. parasiticus*	—↓ aflatoxin production of *A. parasiticus*	[143]

^1^ ARE—antioxidant response element; ^2^ Nrf2—nuclear factor erythroid 2-related factor 2; ^3^ NF-κB—nuclear factor κB; ^4^ HL-60—cells human promyeloid leukemia 60; ^5^ COX-2—cyclooxygenase 2; ^6^ COX-1—cyclooxygenase 1; ^7^ NO—nitric oxide; ^8^ PGE2—prostaglandin E2; ^9^ IFN—γ-gamma interferon; ↓—reduced.

**Table 4 antioxidants-10-00686-t004:** Effects of dietary *Salix* (as bark or leaves) species on heat-stressed broilers.

Species/Origin	Part of Plant and form Used	Dose of Inclusion	Animal Model	Temperature	Effect	Reference
*S. babylonica* (Zai Natural parks, Jordan)	Leaves (extract)	100 mL/day	Arbor Acres broiler chickens	35 °C	—↑ final body weight, average daily gain, and average daily feed intake —↓ feed conversion ratio —↓ rectal temperature —↓ panting rate (breath/minute) —↓ mortality (%)	[153]
*S. alba* (Plant Extract, Radaia, Cluj County)	Bark (hydroalcoholic extract)	0.025% and 0.05% in diet	Cobb 500 broiler chickens	32 °C	—↓ the serum cholesterol, triglycerides, and ALT —↓ the malondialdehyde concentration in liver —↓ the number of *E. coli* and staphylococci in the caecum —↑ the number of lactobacilli in the caecum	[95]
*S. alba* (Plant Extract, Radaia, Cluj County)	Bark (hydroglyceroalcoholic extract)	1% in diet	Cobb 500 broiler chickens	32 °C	—↓ serum cholesterol and glucose level —↓ the pathogenic bacteria in the caecum	[154]
*S. tetrasperma Roxb* (unknown origin)	Leaves (extract)	50 and 100 mg/L of drinking water	No data	34 °C	—no effect on performance —↑ the weight of abdominal fat	[155]

↓—reduced; ↑—increased.

## Data Availability

No new data were created or analyzed in this study. Data sharing does not apply to this article.

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
