# Peer review of "Heat Stress in Broiler Chickens and the Effect of Dietary Polyphenols, with Special Reference to Willow (Salix spp.) Bark Supplements—A Review"

_antioxidants, 2021, doi:10.3390/antiox10050686_

Round 1

Reviewer 1 Report

The Salix metabolites (i.e. extractives and polyphenol colorants) are claimed to possess such antibacterial properties that promote growth of food animals by suppressing the multidrug-resistant strains of zoonotic bacteria. This has been confirmed to some extent scientifically from many individual scientific publications, but in order to gain a comprehensive knowledge about the significant role of the underappreciated willow bark to the poultry’s health, we need a systematic review summarizing the mechanism effect of dietary willow polyphenols to counteract the heat stress in poultry industry. 
The authors have given a nice review about the relationship between heat stress and oxidative stress in broilers, effect of hybrids and season on its chemical composition of Salix metabolites, biological effects of Salix extract in different in-vitro experimental models and the broiler’s performance characteristics that associated with in-vivo feeding trials. However, the authors also need pay attention to other critical aspects listed below to improve the overall quality.
1)    The dietary alternative growth promoters are used to minimize the poultry growth inhibition caused by heat stress, to gain profitable production for the poultry industry. Although the mechanism of growth promotion in food animals is not fully established, antimicrobial agents are presumed to maintain intestinal health by regulating bacterial populations of the microbiome. Use of traditional antibiotics causes the resistance of pathogen bacteria, which subsequently pose a health threat to human – for example, campylobacter and salmonella pathogens, both resistant to quinolones, can infect animals and humans. European Antimicrobial Resistance Surveillance Network reported that antimicrobial resistance infections have been responsible for 33,000 annual deaths by 2015. Nevertheless, the global consumption of the antimicrobials in food animals is predicted to increase by 67% from 63,151 tons (year 2010) to 105,596 tons (year 2030). Hence, there has been a great interest to develop novel alternative growth promoters to hinder the further growth of the antimicrobial resistance in animals and humans. Lines 45-46, the authors claimed the demand for developing the willow metabolites as an alternative to the synthetic pharmaceuticals in poultry production is high. The authors are encouraged to give a summary about the reported consequences of antimicrobial resistance along the food chain at ‘1. introduction’ chapter to strengthen the importance of developing natural metabolites to replace the traditional antibiotics.

Several references are suggested here.

EU (2005) Ban on antibiotics as growth promoters in animal feed enters into effect. Retrieved from URI: https://ec.europa.eu/commission/presscorner/detail/en/IP_05_1687.

ECDC (2018) 33000 people die every year due to infections with antibiotic-resistant bacteria. Retrieved from URI: https://www.ecdc.europa.eu/en/news-events/33000-people-die-every-year-due-infections-antibiotic-resistant-bacteria.

Van Boeckel, Brower, Gilbert, Grenfell, Levin, Robinson, Teillant, Laxminarayan, Global trends in antimicrobial use in food animals. Proc. Natl. Acad. Sci. U.S.A. 112 (2015) 5649. DOI: 10.1073/pnas.1503141112.

2)    We have learned that the stability of the polyphenol compounds, e.g. (+)-catechin (Table 1, Line 244), is poor even under neutral pH, though it is rather stable under acidic environment. Hence, in Chapter 3 the authors are suggested to give a systematic study to uncover the effect of pH and temperature into stability of several selected polyphenols listed at Table 1 (line 244). Because this knowledge is needed to maximize the full gut modulation of pH sensitive molecules and to prevent any degradation under the intestinal acidic environment, when used as dietary additives after feeding purified Salix metabolites into the chickens. One example reference is suggested here to consider.

Zhu, Zhang, Tsang, Huang, Chen, Stability of Green Tea Catechins. J. Agric. Food Chem. 45 (1997) 4624. DOI: 10.1021/jf9706080.

3)    It’s also well known that the microorganisms inhabiting in the gut, play a prominent role in digestion and absorption of the nutrients, thus having strongest influence on the feed efficiency of the food animals. The bacterial metabolism in the gut produces mainly non-digestible carbohydrates, i.e. short-chain fatty acids (SCFA) (e.g. acetate, and butyrate) and the lactic acid. The SCFA compounds can be beneficial to the overall gut health, while lactic acid bacteria inhabit at the small-intestine of chicken, where most of feed-derived energy is absorbed for animal growth. Thus, lactic acid bacteria compete for specific nutrient absorptions directly with the gut host. Hence, their inhibition can significantly enhance the nutrient capture by the host, thereby improving the body weight gain and the feed conversion efficiency. Additionally, those G+ (e.g. Clostridium perfringens and Enterococcus spp.) and G− (e.g. Campylobacter, Salmonella, and E. coli) bacterial strains, which have been reported and investigated as important bird pathogenic bacteria and zoonotic pathogens.

At Table 4 (line 588, Chapter 6) the authors focused on the modulation effects of dietary Salix extracts into the broiler’s performance, such as their weight gain and feed conversion efficiency. However, the microbiota’s modulation, such as their non-digestible carbohydrates composition and those important bird pathogenic bacteria and zoonotic pathogens changes, are also considered as other key criteria in validating the Salix metabolites as alternative growth promoters for food animal. The authors are suggested to make a detailed summary about the modulation response of their microbiota population after feeding purified Salix metabolites as the dietary additives into the chickens. Several literature examples are suggested here to consider.

Ranjitkar, Engberg, The influence of feeding crimped kernel maize silage on growth performance and intestinal colonization with Campylobacter jejuni of broilers. Avian Pathol. 45 (2016) 253. DOI: 10.1080/03079457.2016.1146821.

Ranjitkar, Lawley, Tannock, Engberg, Bacterial Succession in the Broiler Gastrointestinal Tract. Appl. Environ. Microbiol. 82 (2016) 2399. DOI:10.1128/AEM.02549-15.

In addition, the authors also need to clarify these observations
Redundance of ‘the concentration of’ at line 256, please clarify.
1) Table 1, line 248 and line 250, it is confusing that the author shows t= 60°C, are they referring to temperature or extraction period? please clarify.
2) Table 1, Line 250 (ref 86), if the harvesting season for S. aegyptiaca is unknown, please mark it unknown instead of writing like ‘harvest season’.
3) Table 1, literature 87 (Pohjamo et al. 2003) presented the chemical compositions from willow wood and knots, not discussing about the bark section. Additionally, it is S. caprea hybrid not S. daphnoides. please clarify.
4) Table 2 (literature 95), you can’t specify the sample’s origin as ‘market product’, you need explain its origin carefully, e.g. which country and where it was harvested at which season ect. There are many similar descriptions throughout the entire manuscript, please clarify them.

Author Response

Response to Reviewer 1 Comments

Dear reviewer,

Please find our point-to-point response to your useful observations/suggestions.

Point 1:  1)    The dietary alternative growth promoters are used to minimize the poultry growth inhibition caused by heat stress, to gain profitable production for the poultry industry. Although the mechanism of growth promotion in food animals is not fully established, antimicrobial agents are presumed to maintain intestinal health by regulating bacterial populations of the microbiome. Use of traditional antibiotics causes the resistance of pathogen bacteria, which subsequently pose a health threat to human – for example, campylobacter and salmonella pathogens, both resistant to quinolones, can infect animals and humans. European Antimicrobial Resistance Surveillance Network reported that antimicrobial resistance infections have been responsible for 33,000 annual deaths by 2015. Nevertheless, the global consumption of the antimicrobials in food animals is predicted to increase by 67% from 63,151 tons (year 2010) to 105,596 tons (year 2030). Hence, there has been a great interest to develop novel alternative growth promoters to hinder the further growth of the antimicrobial resistance in animals and humans. Lines 45-46, the authors claimed the demand for developing the willow metabolites as an alternative to the synthetic pharmaceuticals in poultry production is high. The authors are encouraged to give a summary about the reported consequences of antimicrobial resistance along the food chain at ‘1. introduction’ chapter to strengthen the importance of developing natural metabolites to replace the traditional antibiotics.

Several references are suggested here.

EU (2005) Ban on antibiotics as growth promoters in animal feed enters into effect. Retrieved from URI: https://ec.europa.eu/commission/presscorner/detail/en/IP_05_1687.

ECDC (2018) 33000 people die every year due to infections with antibiotic-resistant bacteria. Retrieved from URI: https://www.ecdc.europa.eu/en/news-events/33000-people-die-every-year-due-infections-antibiotic-resistant-bacteria.

Van Boeckel, Brower, Gilbert, Grenfell, Levin, Robinson, Teillant, Laxminarayan, Global trends in antimicrobial use in food animals. Proc. Natl. Acad. Sci. U.S.A. 112 (2015) 5649. DOI: 10.1073/pnas.1503141112.

Response 1: According to reviewer, there is a need to strengthen the importance of developing natural metabolites to replace the traditional antibiotics in the introduction section. We revised the introduction and we added the idea suggested by reviewer as follows: For a long time, traditional antibiotics have been administered intensively to animals to prevent disease or as growth promoters (Van Boeckel et al., 2015), with some experts esti-mating that global consumption of antimicrobials in animals is twice that of humans (Aarestrup, 2012). But this intensive use has led to the resistance of pathogenic bacteria, with significant public health implications such as bacterial infections with quinolones resistant bacteria (Campylobacter and Salmonella). Based on this concern, the EU Commission (2005) decided to ban the use of antibiotics as growth promoters in feed. Nevertheless, according to Van Boeckel et al., (2015), the global consumption of the antimicrobials in food animals (2010-2030) is expected to increase by 67%. Hence, there has been a great in-terest to develop novel alternative growth promoters to hinder the further growth of the antimicrobial resistance in animals and humans and to minimize the poultry growth in-hibition caused by heat stress, to gain profitable production for the poultry industry.  ’’(lines 51-62).

Point 2: 2)    We have learned that the stability of the polyphenol compounds, e.g. (+)-catechin (Table 1, Line 244), is poor even under neutral pH, though it is rather stable under acidic environment. Hence, in Chapter 3 the authors are suggested to give a systematic study to uncover the effect of pH and temperature into stability of several selected polyphenols listed at Table 1 (line 244). Because this knowledge is needed to maximize the full gut modulation of pH sensitive molecules and to prevent any degradation under the intestinal acidic environment, when used as dietary additives after feeding purified Salix metabolites into the chickens. One example reference is suggested here to consider.

Zhu, Zhang, Tsang, Huang, Chen, Stability of Green Tea Catechins. J. Agric. Food Chem. 45 (1997) 4624. DOI: 10.1021/jf9706080.

Response 2: According to reviewer, the pH and temperature stability of Salix polyphenols are important to be clearly stated in order to maximize its gut modulation. Thus, we agree with the observation made and consequently we have included some studies: ,,However, we believe that modern strategies for the administration of antioxidants with phenolic structures are needed, due to the different stability of these compounds in acidic, basic or neutral environments. Thus, studies performed on green tea catechins (GTCs), which include (-) - epicatechin (EC), (-) - epicatechin gallate (ECG), (-) - epigallocatechin (EGC) and (-) - epigallocatechin gallate (EGCG ) demonstrated that after oral administration of GTCs, polyphenols were partially absorbed due to the instability in the low alkaline environment of the intestine of EGCG and EGC (Zhu et al., 1997). A promising method of eluting the mechanisms of degradation of polyphenols in alkaline media is the use of co-encapsulation of these compounds in suitable gels. For example, co-encapsulation of EGCG in an emulsion gel containing sucrose and gel in the aqueous phase and polyglycerol polyricinoleate in the oil phase enhanced EGCG chemical stability under simulated gastrointestinal conditions and doubled its bioaccessibility (Chen et al., 2018)’’ (lines 348-360).

Point 3:    3)    It’s also well known that the microorganisms inhabiting in the gut, play a prominent role in digestion and absorption of the nutrients, thus having strongest influence on the feed efficiency of the food animals. The bacterial metabolism in the gut produces mainly non-digestible carbohydrates, i.e. short-chain fatty acids (SCFA) (e.g. acetate, and butyrate) and the lactic acid. The SCFA compounds can be beneficial to the overall gut health, while lactic acid bacteria inhabit at the small-intestine of chicken, where most of feed-derived energy is absorbed for animal growth. Thus, lactic acid bacteria compete for specific nutrient absorptions directly with the gut host. Hence, their inhibition can significantly enhance the nutrient capture by the host, thereby improving the body weight gain and the feed conversion efficiency. Additionally, those G+ (e.g. Clostridium perfringens and Enterococcus spp.) and G− (e.g. Campylobacter, Salmonella, and E. coli) bacterial strains, which have been reported and investigated as important bird pathogenic bacteria and zoonotic pathogens.

At Table 4 (line 588, Chapter 6) the authors focused on the modulation effects of dietary Salix extracts into the broiler’s performance, such as their weight gain and feed conversion efficiency. However, the microbiota’s modulation, such as their non-digestible carbohydrates composition and those important bird pathogenic bacteria and zoonotic pathogens changes, are also considered as other key criteria in validating the Salix metabolites as alternative growth promoters for food animal. The authors are suggested to make a detailed summary about the modulation response of their microbiota population after feeding purified Salix metabolites as the dietary additives into the chickens. Several literature examples are suggested here to consider.

Ranjitkar, Engberg, The influence of feeding crimped kernel maize silage on growth performance and intestinal colonization with Campylobacter jejuni of broilers. Avian Pathol. 45 (2016) 253. DOI: 10.1080/03079457.2016.1146821.

Ranjitkar, Lawley, Tannock, Engberg, Bacterial Succession in the Broiler Gastrointestinal Tract. Appl. Environ. Microbiol. 82 (2016) 2399. DOI:10.1128/AEM.02549-15.

Response 3: Given the reviewer suggestion, we included a short scientific comment about the modulation response of microbiota after feeding polyphenols as Salix metabolites, following our scope- the effect of dietary polyphenols from Salix bark in heat- stressed broilers. We found that there is a reciprocal interaction between gut microbiota and polyphenols, and based on this interaction, the growth performance of animals can be improved. The detailed comment was included in the article (lines 632-677).

Although a large number of papers have been published on this topic, the clear relationship between polyphenols- gut microbiota- growth performance of chickens has not yet been fully clarified. Further investigation would be essential to elucidate the interaction between Salix metabolites in the diet, intestinal microbiota and growth performance, clarifying those genes and microorganisms involved in the metabolization of polyphenols.

Point 4: In addition, the authors also need to clarify these observations

Redundance of ‘the concentration of’ at line 256, please clarify.

Response 4: We revised the sentence expression (line 264).

Point 5: 1) Table 1, line 248 and line 250, it is confusing that the author shows t= 60°C, are they referring to temperature or extraction period? please clarify.

Response 5: We deleted,, t=’’. The value is referring to temperature parameter (table 1 continued ref 88) .

Point 6: 2) Table 1, Line 250 (ref 86), if the harvesting season for S. aegyptiaca is unknown, please mark it unknown instead of writing like ‘harvest season’.

Response 6: According to reviewer suggestion we modified the term,, harvested season ‘’ with ,, unknown season ’’ (table 1 continued, ref 89,90; table 2 ref 93, 95, 96;  table 2 continued, ref 97,89,98).

Point 7: 3) Table 1, literature 87 (Pohjamo et al. 2003) presented the chemical compositions from willow wood and knots, not discussing about the bark section. Additionally, it is S. caprea hybrid not S. daphnoides. please clarify.

Response 7: Unfortunately, it was a mistake in the citation; it is Kamerer et al., (2005) instead of Pohjamo et al., (2013). We modified both in the table, in the text and in the reference section (table 1 and literature 90).

Point 8: 4) Table 2 (literature 95), you can’t specify the sample’s origin as ‘market product’, you need explain its origin carefully, e.g. which country and where it was harvested at which season ect. There are many similar descriptions throughout the entire manuscript, please clarify them.

Response 8: According to reviewer suggestion we clarified the origin of Salix sample in the entire manuscript (table 1 continued, ref.90; table 2, ref 95 and 96; table 2 continued ref 98; table 4 ref 95,154).

Reviewer 2 Report

The work under the title “Heat stress in broiler chickens and the effect of dietary polyphenols, with special references to willow (Salix spp) bark supplements - A review” presents an interesting overview of Salix spp. chemical characterization in terms of its bioactive compounds with their application in the nutrition of broilers exposed to heat stress. The work is well documented, structured and written. Thus, I recommend the publication of this paper with some minor corrections.

Comments

Table 1.

Column 3. It is not necessary to put full description. It is enough to mentioned maceration with time and solvent used for extraction, without filtration, evaporation.

Column 4. HPLC with diode array detection instead HPLC method with HPLC-diode array detection system equipped with an autosampler

Column 5.

Due to the fact that mass fraction of all identified phenolic compounds is expressed in mg/g it is better to insert the mass fraction of polyphenols in mg/g instead polyphenols (rows).

See also column 6 for saligenin derivatives.

Table 1. (continued)

Columns 5 and 6. If the phenolic compounds and saligenin derivates are quantified some measured values must be put. Otherwise the authors should use the term identified. See examples for catechin, 1,2-benzenedicarboxylic acid, etc.

Ethanolic extract (1:10, w/v) and 70% ethanol are the solvents used for extraction. The methods of extraction are not inserted.

See also Table 2. (column 3). Column 4. Under the title total phenolic the authors should put mg GAE/g. See also column 5. See also Table 2 (continued).

In my opinion the Tables should be modify to avoid the writing of the same units, etc. In addition, the most important information must be insert. The others auxiliary reagents, techniques, etc. could be found in references.

Author Response

Response to Reviewer 2 Comments

Dear reviewer,

Please find our point-to-point response to your useful observations/suggestions.

Point 1: Table 1.

Column 3. It is not necessary to put full description. It is enough to mentioned maceration with time and solvent used for extraction, without filtration, evaporation

Response 1: According to reviewer observation we deleted the supplementary operations of extraction (table 1, table 1 continued, table 2, table 2 continued)

Point 2: Column 4. HPLC with diode array detection instead HPLC method with HPLC-diode array detection system equipped with an autosampler

Response 2: According to reviewer observation, we modified in the table (table 1, column 4, table 1 continued)

Point 3: Column 5.

Due to the fact that mass fraction of all identified phenolic compounds is expressed in mg/g it is better to insert the mass fraction of polyphenols in mg/g instead polyphenols (rows).

See also column 6 for saligenin derivatives.

Response 3: According to reviewer observation we deleted the unit in rows (mg/g) and we insert it in the title of columns 5 and 6. Because we have both quantified and identified polyphenols, we inserted the term,, identified’’ in the title of columns 5 and 6 (table1 line 255; table 1 continued, line 258; table 1 continued, line 261).

Point 4: Table 1. (continued)

Columns 5 and 6. If the phenolic compounds and saligenin derivates are quantified some measured values must be put. Otherwise the authors should use the term identified. See examples for catechin, 1,2-benzenedicarboxylic acid, etc.

Response 4: Because we have both quantified and identified polyphenols, we inserted the term,, identified’’.

Point 5: Ethanolic extract (1:10, w/v) and 70% ethanol are the solvents used for extraction. The methods of extraction are not inserted.

Response 5: According to reviewer observation, we inserted the extraction methods (table 1 continued).

Point 6: See also Table 2. (column 3). Column 4. Under the title total phenolic the authors should put mg GAE/g. See also column 5. See also Table 2 (continued).

Response 6: According to reviewer observation, we inserted the extraction methods (table 2 column 3, table 2 continued column 3). We inserted the unit (mg GAE/g) in the title of column 4 and 5 and we deleted it from the rows (table 1 continued). The same modifications were made for table 2, table 2 continued. We deleted mg GAE/g from the rows of column 4 and mg QE/g dw from the rows of column 5 and we inserted it in the title of corresponding column (table 2 and table 2 continued column 4, 5).

Point 7: In my opinion the Tables should be modify to avoid the writing of the same units, etc. In addition, the most important information must be insert. The others auxiliary reagents, techniques, etc. could be found in references.

Response 7: We agree the reviewer suggestions and we modified accordingly (table 1, table 1 continued, table 2, table 2 continued).

Round 2

Reviewer 1 Report

The authors have addressed the comments properly. I would recommend to publish this article.